# The Glucocorticoid Receptor in Cardiovascular Health and Disease

**DOI:** 10.3390/cells8101227

**Published:** 2019-10-09

**Authors:** Bing Liu, Tie-Ning Zhang, Jessica K. Knight, Julie E. Goodwin

**Affiliations:** 1Department of Pediatrics, Yale University School of Medicine, New Haven, CT 06520, USA; bing.liu@yale.edu (B.L.); tie-ning.zhang@yale.edu (T.-N.Z.); Jkk37@cam.ac.uk (J.K.K.); 2Vascular Biology and Therapeutics Program, Yale University School of Medicine, New Haven, CT 06520, USA; 3Department of Pediatrics, Shengjing Hospital of China Medical University, Shenyang 110004, China

**Keywords:** cardiovascular disease, glucocorticoid receptor, circadian rhythm, heart failure, blood pressure, sepsis, nuclear hormone

## Abstract

The glucocorticoid receptor is a member of the nuclear receptor family that controls many distinct gene networks, governing various aspects of development, metabolism, inflammation, and the stress response, as well as other key biological processes in the cardiovascular system. Recently, research in both animal models and humans has begun to unravel the profound complexity of glucocorticoid signaling and convincingly demonstrates that the glucocorticoid receptor has direct effects on the heart and vessels in vivo and in vitro. This research has contributed directly to improving therapeutic strategies in human disease. The glucocorticoid receptor is activated either by the endogenous steroid hormone cortisol or by exogenous glucocorticoids and acts within the cardiovascular system via both genomic and non-genomic pathways. Polymorphisms of the glucocorticoid receptor are also reported to influence the progress and prognosis of cardiovascular disease. In this review, we provide an update on glucocorticoid signaling and highlight the critical role of this signaling in both physiological and pathological conditions of the cardiovascular system. With increasing in-depth understanding of glucocorticoid signaling, the future is promising for the development of targeted glucocorticoid treatments and improved clinical outcomes.

## 1. Introduction

Cardiovascular diseases (CVDs) are disorders of the heart and blood vessels and are the leading cause of death globally. An estimated 17.9 million people died from CVDs in 2016, representing 31% of all deaths worldwide [1]. CVDs are a diverse group of conditions, which include such entities as coronary artery disease, stroke, peripheral vascular disease, rheumatic heart disease, and pulmonary embolism. These diseases may occur throughout the lifespan, however, the vast majority of all deaths due to CVD are the result of one of six conditions: ischemic heart disease, stroke, hypertensive heart disease, cardiomyopathy, rheumatic heart disease, and atrial fibrillation [2].

Common behavioral risk factors for CVDs include unhealthy dietary patterns, physical inactivity, high sodium/tobacco consumption, and alcohol misuse. In addition to these risk factors, several quantifiable physiological changes are associated with CVD, such as high blood pressure (hypertension), elevated blood glucose (diabetes mellitus), elevated cholesterol (hyperlipidemia), being overweight, and obesity. Genetic predisposition and family history of CVDs [3,4,5] are considered risk factors as well.

The pathological mechanisms underlying CVDs are extremely complicated and involve interactions between diverse bio-molecules. Glucocorticoids and their actions via the glucocorticoid receptor (GR) are highly involved in the genesis and development of CVDs. Glucocorticoids are steroid hormones that are essential for life, which are synthesized and released by the adrenal cortex, under regulation by the hypothalamic-pituitary-adrenal gland axis. Glucocorticoid release occurs in a circadian manner, as well as in response to stress, and coordinates a variety of fundamental processes including the inflammatory and immune responses, metabolic homeostasis, cell proliferation, reproduction, and cognition [6,7].

In addition to their physiological roles, glucocorticoids can influence the cardiovascular system in various pathological conditions [8,9,10]. Synthetic glucocorticoids are commonly prescribed in diverse cardiovascular disorders, including infectious conditions such as rheumatic fever and myocarditis, structural conditions such as conduction defects and cardiomyopathy, and vascular conditions such as angina and acute myocardial infarction [11]. However, due to the existence of severe side effects in many organ systems, the therapeutic benefits of (synthetic) glucocorticoids are limited. These adverse effects include diabetes, abdominal obesity, and hypertension, all of which are risk factors for CVDs. Thus, in order to enhance the safety and efficacy of glucocorticoid treatment and facilitate the development of novel glucocorticoids, it is essential to have comprehensive insight into the specific functions of glucocorticoids in the cardiovascular system. Since binding to the glucocorticoid receptor is a critical process in glucocorticoid action, this review aims to deliver an update on GR signaling and highlight the roles of GR signaling in both physiological and pathological conditions of the cardiovascular system.

## 2. GR Isoforms and Structure

GR is encoded by the gene NR3C1, which is located on chromosome 5q31-32 in humans, and acts as a ligand-inducible transcription factor belonging to the nuclear receptor superfamily [12]. The molecular structure of GR includes four components: (1) an N-terminal transactivation domain (NTD), (2) a central DNA binding domain (DBD), (3) a C-terminal ligand-binding domain (LBD), and (4) a hinge region separating the DBD and the LBD [13]. The NTD is the most variable region among the nuclear receptor superfamily members and contains regulatory regions that allow binding to diverse co-regulators and components of the transcriptional machinery through activation function domain 1 (AF-1). Besides this, it is also the primary site for post-translational modifications. Unlike the NTD, the DBD is the most conserved region among the nuclear receptors and comprises two zinc finger motifs that mediate dimerization and direct binding to specific genomic sequences, known as glucocorticoid response elements (GREs). Meanwhile, the LBD is a highly structured domain, consisting of 12 α-helices and four small β-strands, which forms a hydrophobic cavity for glucocorticoid binding and also contains an activation function (AF2) [14]. Upon ligand binding, the AF2 would experience conformational changes in order to allow the LBD to interact correspondingly with either coactivators or corepressors containing LXXLL motifs. Although AF2 is obligately ligand-dependent, AF1 is constitutively active and can function in the absence of the LBD. GR AF1 has been classified as an intrinsically disordered protein (IDP) and it has been proposed that AF1 can “sample” its environment for appropriate binding partners and result in a rapidly changing molecular conformation [15].

Two nuclear localization signals, NL1 and NL2, are located at the DBD/hinge region and the LBD and are responsible for triggering translocation to the nucleus via an importin-dependent mechanism [16].

The human GR gene consists of nine exons numbered from 1 to 9 (Figure 1). Exon 1 forms the 5′-untranslated region and exon 2 encodes for the entire NTD. Exons 3 and 4 comprise the DBD and exons 5–9 comprise the LBD. GR has several receptor isoforms, for example, GRα and GRβ, which differ only in their C-termini and result from splicing differences in exon 9 [17,18] (Figure 1). The GRα isoform undergoes alternative translation initiation from conserved AUG start codons in exon 2 of the GR gene, generating eight additional isoforms of GR (GRα-A, GRα-B, GRα-C1, GRα-C2, GRα-C3, GRα-D1, GRα-D2, and GRα-D3) with progressively shorter NTDs [19]. Although they do demonstrate similar binding affinities for glucocorticoids and a similar ability to interact with GREs [19], there are some isoform-specific differences. For instance, the GR-C isoform was demonstrated to confer an increased susceptibility to apoptosis in cell culture while the GR-D isoform resulted in a relative resistance to apoptosis under the same conditions [20].

The GR primary transcript consists of nine exons and GR protein comprises NTD, DBD, “hinge region”, and LBD. The location of AF1, AF2, NL1, and NL2 are shown. Common GR isoforms like GRα, GRβ, GRγ, GR-A, and GR-P are generated after specific alternative splicing.

Unlike GRα, the GRβ splice variant does not bind glucocorticoids and resides constitutively in the nucleus of cells, acting as a natural dominant negative inhibitor of the GRα isoform on many glucocorticoid-responsive target genes [21]. Increased expression of GRβ has been associated with glucocorticoid resistance, which may be due to competition for GRE binding, competition for transcriptional co-regulators, or the formation of inactive GRα/GRβ heterodimers. The ability of GRβ to inhibit the activity of GRα suggests that high levels of GRβ may lead to glucocorticoid resistance [22]. Furthermore, GRβ may also generate eight β isoforms that are similar to the GRα [17] and is also able to exert its own independent functions, which include suppressing the transcriptional activity of the GATA3 transcription factor on its responsive IL-5 and -13 promoters by attracting histone deacetylases [23]. There is also evidence that GRβ has some GRα-independent transcriptional activity, as Kino et al. showed in HeLa cells [24].

Additional GR isoforms have been associated with glucocorticoid insensitivity [16]. GRγ exhibits about 50% of the activity of GRα for canonical glucocorticoid target genes. Recently, the level of GRγ was correlated with glucocorticoid resistance in childhood acute lymphoblastic leukemia and small cell lung carcinoma [25,26]. Two others, GR-P and GR-A, are splice variants that are missing large regions of the LBD (GR-P: missing appropriate exons 8 and 9, GR-A: missing the entire sequences of exons 5, 6, and 7) [27]. Due to these defective changes in the LBD, GR-P and GR-A do not bind glucocorticoids and both GR-P and GR-A have been determined to contribute to glucocorticoid resistance. Of note, GR-P is found in many tissues and has been shown to modulate the transcriptional activity of GRα in a cell-specific manner [17,28,29].

Specific tissues display particular relative abundances of these GR isoforms and react in individualized fashions to unique stimuli such as differentiation and aging. These differences are of crucial importance in determining the tissue-specific actions of glucocorticoids.

## 3. Genomic and Non-Genomic Effects of GR

### 3.1. Genomic Effects of GR

Classically, the effects of glucocorticoid signaling are genomic, meaning that they are governed by GR-mediated transcription and protein synthesis. In its quiescent state, GR is located in the cytoplasm, which is bound to a chaperone complex which consists of the heat shock proteins hsp90, hsp70, and hsp56, as well as the immunophilins KBP51, FKBP52, Cyp44, and PP5 [17,30]. Upon ligand binding, the GR complex undergoes a conformational change, involving post-translation modifications such as phosphorylation and acetylation. This structural rearrangement exposes the two nuclear localization signals and GR rapidly translocates into the nucleus, where it can exert its actions through genomic (transactivation and transrepression) mechanisms [13]. Within the nucleus, GR regulates the transcription of its target genes in three main ways (Figure 2): (i) direct binding to GREs, (ii) interacting with other transcription factors, or (iii) by both direct binding to GREs and interaction with other transcription factors.

In classical “direct” GR transcriptional regulation, GR that is bound to its ligand homodimerizes in the nucleus and exerts transcriptional activity by direct binding to GREs. The classic GRE sequence, 5′-AGAACAnnnTGTTCT-3′, is a palindromic sequence that is composed of two hexamers that are separated by three base pairs. These bases may be any DNA residue, though one is typically highly conserved, while the other two are more variable [18]. GR binds to GREs as a homodimer, with each half site occupied by one receptor subunit. The three-nucleotide spacing between the two half sites is strictly required for GR to dimerize on directly regulated GREs. Meanwhile “tethering” GREs allow GR to indirectly regulate gene expression, without the receptor itself binding DNA. Although lacking a DNA binding site, tethering GREs can recruit other transcription factors that bind to GR [16]. GR can also regulate gene expression by binding to “composite” GREs. In this case, target genes contain binding sites for both GR and other transcription factors [18].

Historically, GREs have been shown to mediate the glucocorticoid-dependent induction of many genes and have been regarded exclusively as activating. More recently, data have shown that GR occupancy of canonical GREs can result in the repression of target genes as well [31].

In contrast to the regulation of classical GREs, the repression of negatively regulated target genes is usually mediated by negative GREs (nGREs). The consensus nGRE sequence, 5′-CTCC(n)0-2GGAGA-3′, is distinct from the classic GRE in sequence and function. The nGRE sequence has a variable spacer ranging from 0 to 2 nucleotides and (unlike classical GREs) is occupied by two GR monomers that do not homodimerize [32]. nGREs are abundant throughout the genome and contribute to the regulation of the hypothalamic-pituitary-adrenal (HPA) axis, inflammation, and angiogenesis.

Both excesses and deficiencies of glucocorticoids, mediated by changes in genomic GR signaling pathways as described above, can lead to pathological conditions and impair the cardiovascular stress response [33]. For instance, hypotension, hypoglycemia, and pancytopenia are regarded as signs of cortisol insufficiency; however, the provision of large doses of exogenous glucocorticoids may result in hypertension [34] and decreased total peripheral vascular resistance in both healthy patients and those in shock [9]. In addition, excess glucocorticoids also induce pathophysiological changes in the myocardium through the angiotensin II signaling pathway [35,36,37].

### 3.2. Non-Genomic Effects of GR

In addition to functioning through genomic mechanisms, which usually occur in hours, glucocorticoids can also exert their actions more rapidly (within minutes) via non-genomic signaling mechanisms (Figure 2). Instead of requiring nuclear GR-mediated transcription and translation, such non-genomic actions are initiated at the cell surface via either membrane-bound [38] or cytoplasmic GR [39]. During the structural change in the GR–ligand complex that follows glucocorticoid binding, GR-bound proteins including HSPs and Src are released from the multimeric GR complex [40,41,42,43] and these dislocated components from the GR complex can themselves influence cellular signaling (Figure 2). For example, GR has been reported to activate the PI3K-Akt pathway [44], which in turn can activate eNOS and has been shown to reduce myocardial infarct size [45]. Other evidence suggests that GC-GR signaling can induce rapid biological modulation in contractility, vascular reactivity, and blood pressure in a non-genomic manner in the cardiovascular system [9].

## 4. Glucocorticoid Metabolism in the Cardiovascular System

The availability of cellular glucocorticoid is regulated by the tissue-specific metabolic enzymes 11β-hydroxysteroid dehydrogenase 1 and 2 [46]. 11β-HSD2 converts active cortisol into inactive cortisone, while 11β-HSD1 converts cortisone to cortisol (Figure 3). Since the activity of 11β-HSD2 in the vasculature is minimal, the cardiovascular system is directly affected by circulating cortisol levels [17]. It is important to note that the expression of 11β-HSD2 can be up regulated under conditions of chronic, intermittent hypoxia and in this case, the increased 11β-HSD2 level plays an essential role in the regulation of tissue sensitivity to glucocorticoids [9]. Furthermore, circulating glucocorticoid levels can be modulated by corticosteroid binding globulin (CBG) [47], which not only accelerates the distribution of cortisol, but also participates in its release to tissues.

## 5. GR Polymorphisms

In addition to the genomic and non-genomic effects of GR, GR polymorphisms are also reported to influence cardiovascular disease. The capacity of GR to exert activation or repression of transcriptional regulation is affected by several polymorphisms in the GR gene that alter the amino acid sequence of the encoded receptor. Subsequently, such genetic variations can affect both the efficiency of GC therapy and disease pathology [48,49]. Table 1 describes several key pathological mutations that have been identified in the human NR3C1 gene [50,51,52,53,54,55,56,57,58,59,60,61,62,63,64,65,66,67,68,69,70,71]. For cardiovascular disease or its underlying risk factors, the most extensively studied polymorphisms are the ER22/23EK, GR-9R, N363S, and BclI polymorphisms. Generally speaking, glucocorticoid hypersensitivity has been identified when N363S and BclI polymorphisms are present in the context of visceral obesity. However, other polymorphisms, such as ER22/23EK or GR-9β, are associated with glucocorticoid resistance in the context of healthier metabolic syndrome.

For example, the ER22/23EK (rs6189 and rs6190) polymorphism, which occurs in about 3% of the population, is associated with a more favorable metabolic profile, including a lower risk of both type 2 diabetes and cardiovascular disease [50]. This effect is conferred via reduced glucocorticoid sensitivity, which is mediated by a change from GAG AGG to GAA AAG in exon 2 at position 23 within the NTD [72,73].

The N363S (rs6195) polymorphism, located in exon 2, is found at a 4% frequency in the general population. It corresponds to a change from an A to a G [74]. Unlike ER22/23EK, the presence of the N363S polymorphism enhances the transcriptional activity of GR and is associated with glucocorticoid hypersensitivity. Patients carrying this mutation have a higher body mass index [51], as well as an increased incidence of obesity [52,53], type 2 diabetes [54], and coronary artery disease [55].

The GR-9β polymorphism, which occurs in 8% of the general population, consists of an A to G substitution in the 3′ UTR end of exon 9 (GTTTA motif). This change also confers a more favorable metabolic profile and is associated with a decreased risk of obesity in women, decreased total cholesterol levels, and increased HDL cholesterol levels. In addition, a recent clinical study found that the GR-9β polymorphism is also involved in the regulation of blood pressure [56].

Finally, the BclI polymorphism is located in intron 2 with a C to G nucleotide change and is found in 25% of the general population. As mentioned above, it is associated with GC hypersensitivity [57,58]. Many studies have demonstrated that the BclI mutation is related to a wide variety of metabolic, cardiovascular, and psychiatric disorders, such as hypertension, adiposity, obesity [59,60], and bulimia nervosa [61]. Furthermore, BclI carriers also have a higher risk of post-traumatic stress disorder [62], cardiovascular disease [63], and atherosclerosis [64]. There are other known polymorphisms that influence the risk of cardiovascular disease, including D401H, A714Q, F737L, F774S, V575G, D641V, and G679S, and these are also listed in Table 1.

## 6. The Role of GR in Physiological Conditions

### 6.1. GR in Cardiac Development

The surge in levels of maternal glucocorticoids entering the fetal circulation that is observed during late gestation suggests an essential role for glucocorticoids in preparation for life after birth [75]. Studies in vivo have found that globally GR-deficient (GR^-^/^-^) mice are glucocorticoid resistant and all die soon after birth, which implies the critical role of GR in the development of organ systems that are adapted to postnatal survival [76]. Compared with other organs, the heart is functional relatively early in gestation and is required for fetal survival [77]. To reveal the potential role of GR during cardiac development, Oakley et al. developed a mouse model lacking GR expression in heart tissue (the cardiomyocyte-specific GR knock out) [78]. They found that although at first the mice were born in the expected Mendelian ratios and displayed a normal phenotype, they died prematurely (median survival approximately 7 months) from pathological cardiac hypertrophy that progressed to dilated cardiomyopathy and heart failure [78]. Additionally, Rog-Zielinska et al. demonstrated that application of glucocorticoid to primary mouse fetal cardiomyocytes could replicate the structural, functional, and biochemical consequences of GR activation in heart tissue [77]. This includes improving the contractility of cardiomyocytes, promoting Z disc assembly and the appearance of mature myofibrils, and increasing mitochondrial activity [77]. These effects demonstrate the potentially vital role of GR in intrauterine cardiac development.

In addition to studies in vivo, there are also several in vitro studies that demonstrate an important role for GR during heart development. In order to understand the influence of GR at the gene level, Oakley et al. constructed the GR^-^/^-^ mouse and found that the expression of several key genes changed relative to wild type, including some critical genes for cardiac contractility (ryanodine receptors 2, RyR2), cardiomyocyte survival (prostaglandin D2 synthase, Ptgds), and inhibition of inflammation (lipocalin 2, Lcn 2) [78]. This suggests that there is specific activation of GR in heart tissue and reveals an obligate role for GR in maintaining normal cardiovascular function. At the molecular level, another recent study identified genes that are directly regulated by GR using next-generation sequencing. In this work, the structural maturation of fetal cardiomyocytes catalyzed by glucocorticoid treatment was shown to be critically influenced by the transcriptional coactivator, PGC-1α [79].

Similar investigations have also been performed in non-mouse animal models. For example, Kim et al. found that developing piglet hearts had smaller myocytes, with reduced binucleation, fewer apoptotic nuclei, and more proliferative nuclei than term hearts before the glucocorticoid surge [80]. This suggests that GR-related pathways could be involved in the regulation of myocyte size during cardiac development. An additional study performed by Eiby et al. showed that glucocorticoids could induce myocyte structural maturation through GR-related pathways and thus, influence heart function [81]. They isolated working hearts from piglets and stabilized them in the Langendorff mode to demonstrate the influence of GR on myocyte structure [81]. From these in vitro and in vivo studies, which are summarized in Table 2, it is evident that glucocorticoids and GR are critical for cardiac development and maturation.

Many clinical trials have suggested that sustained exposure or repeated low doses of glucocorticoids prenatally in humans are effective in improving fetal outcomes [82,83]. Studies have found that exposure to multiple courses of prenatal corticosteroid therapy resulted in a significant reduction in neonatal morbidity [83] and especially in the incidence of respiratory distress syndrome [82]. However, specific data regarding the effect of prenatal glucocorticoids on human cardiac development and maturation are still lacking. Several studies have identified the potential role of GR-related signaling in cardiac development, while a smaller number have demonstrated that excessive prenatal glucocorticoid exposure could potentially result in a delay in cardiac maturation and even cause cardiovascular problems in adult life [84,85]. As such, it is evident that the time- and concentration-dependent effects of GR-related pathways in cardiac development are still not fully understood and require further exploration in future studies. 

### 6.2. GR in Cardiac Contractility

Cardiac contractility is regulated by changes in the concentration of intracellular calcium ([Ca^2+^]_i_) and calcium signaling has a key role in maintaining normal function in myocardial cells. Maintenance of normal heart function requires that cytoplasmic [Ca^2+^]_i_ is sufficiently high in systole and low in diastole [86,87] to allow contraction and relaxation of cardiomyocytes. Under physiological conditions, cytosolic calcium binds to troponin, resulting in sliding of the thick and thin filaments. This causes cell shortening, the development of pressure within the ventricle, and, finally, the ejection of blood [88]. Calcium channels may be divided into three groups: L-type, T-type, and N-type channels. In heart tissue, the L-type calcium channel is the main type in the adult and plays an important role in the process of myocardial contraction. The opening of L-type calcium channels generates depolarization and the entry of a small number of calcium ions, which contributes to a significant increase in [Ca^2+^]_i_ in the dyadic space (the region bounded by the t-tubule and sarcoplasmic reticulum (SR)). This small increase in [Ca^2+^]_i_ then causes the SR calcium release channels to open, releasing a much larger amount of calcium from the SR in a process of positive feedback. For relaxation to occur, calcium is removed from the cytoplasm through RyR closure, then calcium is re-sequestered in the SR, reducing the free cytosolic [Ca^2+^]_i_.

Recently, several studies have found that GR may have essential functions during cardiac contractility both in vivo and in vitro. For example, Oakley et al. generated mice with cardiomyocyte-specific deletion of GR (cardio GR knockout (KO)) and found that these mice displayed different phenotypes, including heart dysfunction, compared with wild type mice [89], thus demonstrating that insufficient GR signaling is pathogenic in cardiomyocytes. To investigate these mechanisms in more depth, Oakley et al. used a microarray technique to examine the signaling pathways that are associated with the dysregulated genes and found that “Cardiac β-Adrenergic Signaling” was strongly associated with the dysregulated genes in the cardio GR KO heart [89]. Notably, the canonical signaling pathways associated with cardiac contractility through calcium channels, suggesting that GR could also affect calcium signaling in cardiomyocytes. Furthermore, Oakley et al. also examined the mRNA expression three key calcium regulators: (1) voltage-gated L-type calcium channel (LTCC), (2) sarcoplasmic/endoplasmic reticulum calcium adenosine triphosphatase 2 (SERCA2), and (3) sodium/calcium exchanger 1 (NCX1), as well as the RyR2 gene [89]. The expression of these four genes was significantly decreased in cardio GR KO mice relative to wild type, suggesting that calcium handling proteins in the β-adrenergic signaling pathway exhibited genotype-specific differences in expression. Further experiments by Cruz-Topete et al. found that, consistent with the differences in phenotype, hearts from cardio GR KO males displayed a more pronounced dysregulation in the expression of genes involved in heart rate regulation, calcium handling, and cardiomyocyte structure and viability [90]. Notably, the RyR2 gene was among the dysregulated genes in male knockout hearts implicated in calcium handling through RNA-seq analyses [90]. These data suggest that the GR regulates cardiac contractility though L-type calcium channels.

However, not limited to L-type calcium channels, studies in vitro have found that GR may also regulate the inner-heart T-type calcium channels, which are expressed in healthy ventricular myocytes during the fetal and perinatal period, but re-express during several pathological conditions in adults. A recent study showed that dexamethasone induced the up regulation of CaV3.2 mRNA in neonatal rat ventricular myocytes, which was mediated by the activation and nuclear translocation of GR [91], further adding to the complex role of GR in the process of regulating cardiac contractility.

### 6.3. GR in Blood Pressure Regulation

High blood pressure is an independent risk factor for many cardiovascular events. Blood pressure is determined by the product of cardiac output and systemic vascular resistance and is regulated by baroreceptors. Glucocorticoids have been confirmed as vital hormones in the regulation of blood pressure (BP) [92] and there is strong evidence that GR is present in both vascular smooth muscle (VSM) [93,94,95] and vascular endothelial cells [94,96,97]. In a previous study by our group, we created a VSM-specific GR KO mouse model. Control and KO mice had similar baseline BP; however, when provided with exogenous dexamethasone, KO mice demonstrated significantly attenuated acute and chronic hypertensive responses. These data suggest that VSM GR is a critical mediator of the hypertensive response in vivo [98]. Additionally, we investigated the association between BP and GR expressed in vascular endothelial cells by developing tissue-specific KO of GR in the vascular endothelium. We found that the KO mice had slightly elevated baseline BP and were relatively resistant to dexamethasone-induced hypertension compared with wild type mice, highlighting the importance of GR in vascular endothelial cells in the adjustment of BP in vivo [99].

In vitro studies also support an important role for GR in the regulation of BP. Investigations in smooth muscle cell culture found that glucocorticoid could increase the expression of angiotensin II type I receptors, resulting in a change in blood pressure [100,101]. Additionally, other studies indicate that GR, as well as the mineralocorticoid receptor (MR), mediates the effect of glucocorticoids on the influx of Na^+^ and/or Ca^2+^ into vascular smooth muscle, altering contractility and, therefore, influencing blood pressure [94,102].

Besides its effects on smooth muscle cells, GR may also regulate blood pressure via alternative pathways in vascular endothelial cells. For instance, through GR-related signaling, glucocorticoids can influence the nitric oxide synthase pathway by decreasing the expression of guanosine triphosphate cyclohydrolase 1 (GTPCH1) mRNA, which is the rate-limiting enzyme in the production of tetrahydrobiopterin (BH4), [103], and in this way, further influence the response of vascular cells to blood pressure in vitro. GR may thus participate in the regulation of blood pressure through several different pathways in different vascular cell types.

### 6.4. GR in the Circadian Rhythm

Living organisms, including humans, have evolved and maintain the highly conserved circadian clock system to adjust the body’s activity to circadian changes in the environment [104,105]. Clock, the “circadian locomotor output cycle kaput”, and its heterodimer partner “brain-muscle-arntlike protein 1” (Bmal1) play an essential role in the establishment of the circadian rhythm and function as internal circadian timekeepers [105]. A previous study found that circulating hormones acting through GR were involved in the maintenance of peripheral circadian clocks. Dysregulation of these clocks is a known risk factor for myocardial infarction [106], suggesting that GR could participate in the regulation of circadian rhythm and thus influence cardiovascular disease. An in vivo study by our group found that in contrast to control mice, mice with tissue specific KO of vascular endothelial GR were able to recover a significant portion of their normal circadian BP rhythm following dexamethasone administration, suggesting that GCs may act partly through the vascular endothelial GR to disrupt the BP rhythm, which is normally maintained by this peripheral circadian clock [99].

The molecular mechanisms that form peripheral circadian clocks are beginning to be understood. Nader et al. showed that the transcription factor clock, a master regulator of circadian rhythm, acetylates GR and represses the transcriptional activity of several glucocorticoid-responsive genes [107]. GR also suppresses the expression of Rev-erbα (Nr1d1), which is also an important component of circadian regulation [108]. These findings provide new insights into the role of GR in the regulation of the circadian rhythm.

## 7. The Role of GR in Cardiovascular Diseases

### 7.1. GR in Heart Failure

Heart failure has become one of the leading causes of death worldwide. It is well known that glucocorticoids can exert both positive and negative effects on the heart. However, the direct role that GR signaling plays in cardiomyocytes is poorly understood. The first study to demonstrate the role of cardiac GR signaling in vivo indicated that GR could have a direct effect on the heart [109]. Overexpression of human GR (hGR) in cardiomyocytes at three times the level of endogenous GR did not lead to a major alteration in cardiac function, but could generate bradycardia and atrioventricular block [109]. This may have been caused by reduced Na^+^ and K^+^ currents and increased L-type calcium currents, calcium transient amplitudes, and SR calcium content, all of which were induced by hGR overexpression [109]. Additionally, several studies in mice with cardiomyocyte-specific deletion of GR and in mice with specific deletion of GR in both vascular smooth muscle cells and cardiomyocytes have demonstrated that reduced GR could lead to pathological consequences in the adult heart [78,110]. These two models found cardiac hypertrophy in adult GR KO heart and up regulation of myosin heavy chain-β, a marker of pathological cardiac remodeling. Mice with cardiomyocyte-specific deletion of GR appeared normal in early life, but developed left ventricular systolic dysfunction at three months of age and died early due to congestive heart failure [78]. Several key genes showed obvious changes, including a decrease in RyR2 and increases in some genes that are associated with inflammation [78].

A few more recent studies also draw similar conclusions. For example, the cardio GR KO mouse model developed by Oakley et al. spontaneously developed cardiac hypertrophy and left ventricular systolic dysfunction, then died prematurely from heart failure [89]. At a molecular level, this model showed reductions in RyR2, Krüppel-like factor 15 (Klf15), and Ptgds. Since this decreased expression occurred before evidence of functional impairment, these proteins are hypothesized to be proximal mediators of heart failure [89]. Krüppel-like factors are zinc finger DNA-binding transcription factors [111] and have been implicated in the regulation of myriad cellular processes [112], including cardiac biology [113,114]. Klf13 and Klf15, in particular, have both been suggested to influence heart function [113,115]. Work by Cruz-Topete et al. showed that Klf13 mRNA and protein levels were significantly decreased in the hearts of cardiomyocyte GR KO mice, suggesting that they are potentially regulated by GR in the heart [116]. Based on next-generation sequencing, Cruz-Topete et al. also revealed that deleting GR in male mouse hearts leads to a profound dysregulation in the expression of genes involved in calcium handling that are implicated in the progression of heart failure [90].

### 7.2. GR in Atherosclerosis

Atherosclerosis is a chronic metabolic disorder resulting from a complex interplay between genes and the environment, however the underlying mechanism remains unclear [117]. Previous studies showed that glucocorticoids could have an effect on regulating lipid metabolism and thus, influence the process of atherosclerosis. This process may be mediated by enzymes 11β-HSD1 and 11β-HSD2. Both isoforms take part in the regulation of lipid metabolism—for example, studies have found that inhibition of 11β-HSD1 slowed atherosclerosis [118,119], while loss of 11β-HSD2 could lead to striking atherogenesis [120]. However, the function of 11β-HSD1 and 11β-HSD2 may be partly attributed to their suppression of GC-mediated MR activation. It is less understood how GR-mediated pathways contribute to atherosclerosis.

To this end, we developed a mouse model with specific GR^-^/^-^ in vascular endothelial cells. When these animals were bred onto an ApoE (-/-) background, they developed a more severe atherosclerotic phenotype compared to ApoE (-/-) controls. These data support an important role of endogenous corticosterone via endothelial GR in reducing vascular inflammation [121]. However, another study performed by Yang et al. found that there was no impact of elevated cholesterol on the expression of GR or on the hypothalamic-pituitary-adrenal axis, measured by a DEX suppression test [122], suggesting the existence of crosstalk between GR and some other signaling pathways. The exact regulatory mechanisms underpinning the relationship between GR and atherosclerosis still need to be studied further.

### 7.3. GR in Sepsis-Induced Cardiovascular Injury

Sepsis is a syndrome of physiological, pathological, and biochemical abnormalities that is caused by an altered systemic host response to infection [123,124,125]. During sepsis, the cardiovascular system may be the target of infection and there may also be injury to vascular endothelial cells, impaired myocardial contractility, and a reduced cardiac ejection fraction [126]. Although it is still controversial whether patients with sepsis should be administered glucocorticoids clinically, GR has been proven to play an important role in sepsis. One of our previous studies found that mice lacking endothelial GR showed significantly increased mortality, more hemodynamic instability, and higher levels of inflammatory cytokines compared with control mice [127]. The presence of endothelial GR is also required for DEX to rescue animals from LPS-induced sepsis in vivo [128], suggesting that GR is a critical regulator during sepsis. An in vitro study by Dschietzig et al. found that relaxin, a peptide belonging to the insulin superfamily, could improve TNF-α-induced endothelial dysfunction, a process which might also be mediated by GR signaling [129]. At a molecular level, relaxin improved endothelial injury by promoting eNOS activity, suppressing endothelin-1 and arginase-II expression, and up-regulating SOD1 via GR and GR-c/EBP-β pathways [129]. This molecular evidence again shows that GR participates in regulatory processes in endothelial cells under inflammatory conditions.

In addition to its vascular effects, GR also plays a critical role in sepsis-induced myocardial dysfunction. For example, Zhang et al. showed that inhibition of GR signaling pathways with the GR antagonist mifepristone increased the levels of basal and LPS-induced proinflammatory cytokines in a rat model [130]. Further cardiac function studies demonstrated that the blockade of GR-related signaling pathways aggravated inflammation-induced cardiac dysfunction. Moreover, in cecal ligation and puncture mouse models, the other major type of in vivo sepsis model, Abraham et al. found that sepsis could alter the expression of GR-α and GR-βisoforms in heart tissue: sepsis decreased GR-α but increased GR-β protein abundance in the heart [131]. These changes might explain the diminished glucocorticoid responsiveness observed in sepsis. However, further studies are still required to explore the role of GR in sepsis-induced cardiovascular injury.

### 7.4. GR in Cardiac Hypoxia/Ischemia Injury

Hypoxia is one of the most important and clinically relevant physiological stresses and results in increased cardiac vulnerability to ischemic injury. Until now, there have been few studies reporting that GR participates in the process of hypoxia in heart tissue. The majority of studies investigating the role of GR in hypoxia conditions have employed in vitro models. For example, a study performed by Xue et al. found that hypoxia could decrease GR expression in hearts from fetal, 3-week-old, and 3-month-old rats, resulting in decreased GR binding to GREs at the AT2R promoter, which was then followed by improved post-ischemia recovery of left ventricular function and avoidance of the fetal hypoxia-induced cardiac ischemic vulnerability [132]. Hypoxia resulting in a long-term change in GR gene expression in the heart in rat models might be caused by hypermethylation of the GR promoter, which results in a decrease in the binding of transcription factors [133]. Additionally, a recent study performed by Martinez et al. found that microRNA (miRNA) could participate in the regulation of GR in response to hypoxia [134]. They reported that hypoxia induces HIF-1α-dependent miR-210 production and mediates GR suppression in H9c2 rat heart cell line [134]. However, the injury caused by hypoxia may be averted. A recent study performed by Zuo et al. showed that an acid polysaccharide fraction of ginseng (AP1) was the most effective fraction at protecting cardiomyocytes from hypoxia [135]. AP1 exerted a protective effect by increasing the expression of both GR and the estrogen receptor, which in turn mediated the activation of the RISK pathway and eNOS-dependent mechanisms to resist reperfusion injury [136]. Although these in vitro studies provide evidence that GR is involved in hypoxia in heart tissues, there is still a need for more studies that explore the role of GR in heart hypoxia in vivo.

In myocardial infarction, GR has also been shown to both mediate pathological processes and act as a determinant of the disease state. For instance, Xu et al. illustrated that DEX could induce the expression of the Bcl-xL gene in mice and have a potentially protective effect, which could be blocked by the GR antagonist mifepristone, suggesting an important role for GR in myocardial infarction [136]. An additional study performed by Xue et al. drew a similar conclusion [137]. They found that the protective effect of DEX was due to its ability to increase GR binding to GC response elements at AT1aR and AT2R promoters, resulting simultaneously in a significant increase in the expression of AT1R and a decrease in AT2R in the heart [137]. However, further clinical studies are required to confirm these conclusions in humans.

## 8. Conclusions

Tissue-specific signaling via the glucocorticoid receptor has profound implications for cardiovascular health and disease. GR mediates a variety of diverse physiological effects in different tissues, using both genomic and non-genomic signaling mechanisms. The diversity of responses observed in different tissues is possible in part due to the presence of tissue-specific GR isoforms, generated through alternative splicing, alternative translation initiation, and post-translational modifications. Physiological changes generated by GR can include alterations in myocardial contractility, vascular tone, endothelial permeability, and arterial blood pressure. GR has also been shown to be intimately involved in the pathogenesis of many common cardiovascular diseases, including heart failure, atherosclerosis, sepsis, and myocardial ischemia (Figure 4). For example, in animal models, both reduction and overexpression of cardiomyocyte GR can disrupt cardiac contractility, while loss of the endothelial GR has been shown to worsen the development of atherosclerotic plaques and reduce hemodynamic stability in septic shock. The use of GR knockdown cell lines and cell-type-specific knockout mice continue to be important experimental approaches in understanding the molecular mechanisms that underlie these tissue-specific effects.

Glucocorticoids are commonly prescribed drugs in various conditions including autoimmune disease and transplant medicine, however their therapeutic utility is often limited by off-target adverse effects. Improved understanding of GR in cardiovascular biology may allow better prediction and prevention of many side effects of glucocorticoid treatment. Simultaneously, understanding the role of endogenous glucocorticoids in the pathogenesis of cardiovascular diseases may permit the development of novel therapeutic strategies that exploit these signaling pathways.

### Future Directions

Targeting endothelial GR-linked pathways in septic shock or atherosclerosis.Modulating cardiomyocyte GR-linked pathways in heart failure or myocardial infarctionInvestigating GR regulation of peripheral circadian rhythms.Controlling tissue microenvironments by 11-β-HSDs.Exploring cardiovascular risk reduction models using GR biology.

Future research into the role of GR signaling in cardiovascular disease thus has the potential to provide both scientific insights and clinical practice improvement.

## Figures and Tables

**Figure 1 cells-08-01227-f001:**
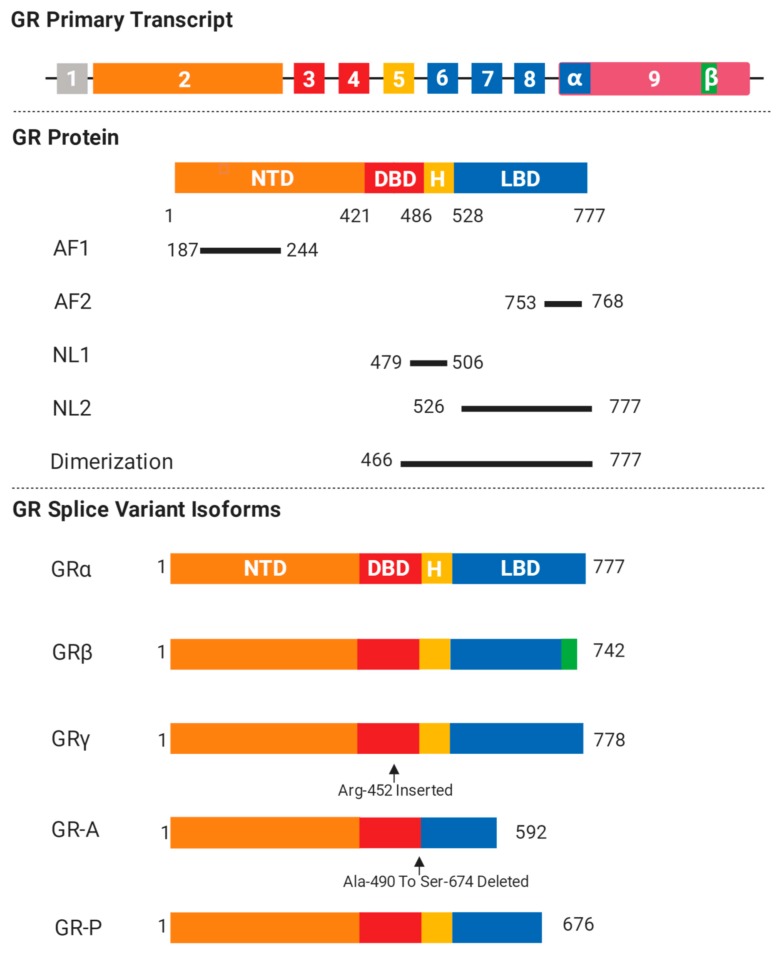
Glucocorticoid receptor (GR) isoforms and structure.

**Figure 2 cells-08-01227-f002:**
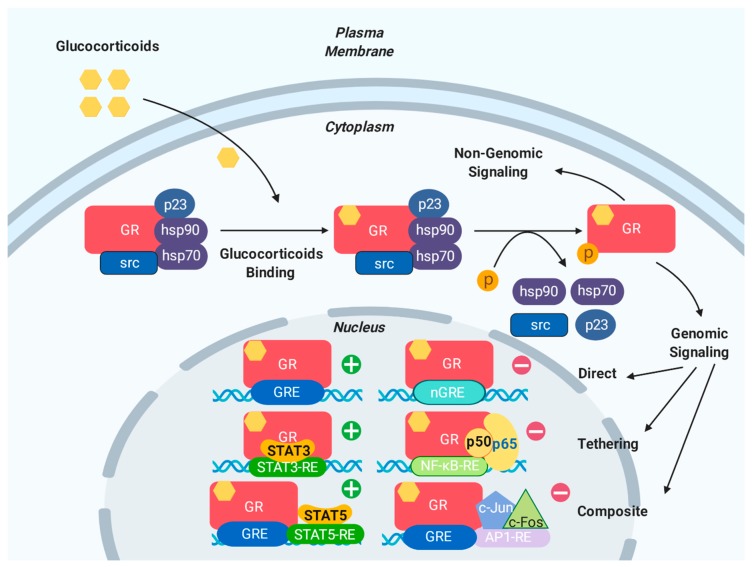
GR signaling pathway. Glucocorticoids bind to GR via both genomic and non-genomic signaling pathways. Once inside the nucleus, GR can function in three ways, all of which can activate or repress gene expression.

**Figure 3 cells-08-01227-f003:**
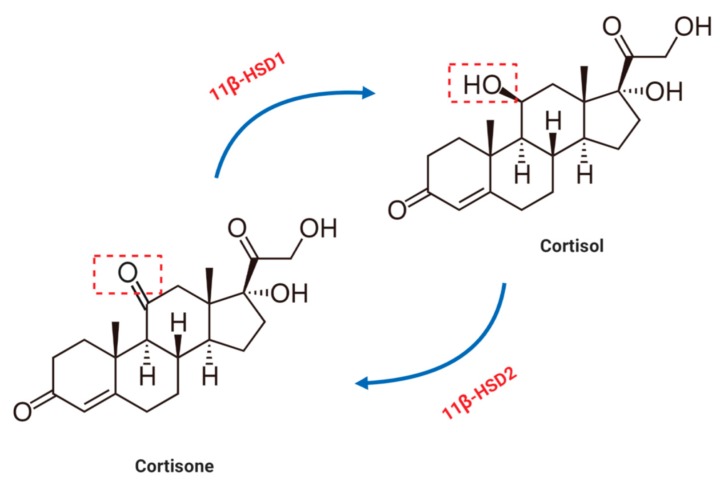
Metabolism of endogenous steroid. 11β-HSD1 and 11β-HSD2 regulate the conversion between cortisol and cortisone in vivo.

**Figure 4 cells-08-01227-f004:**
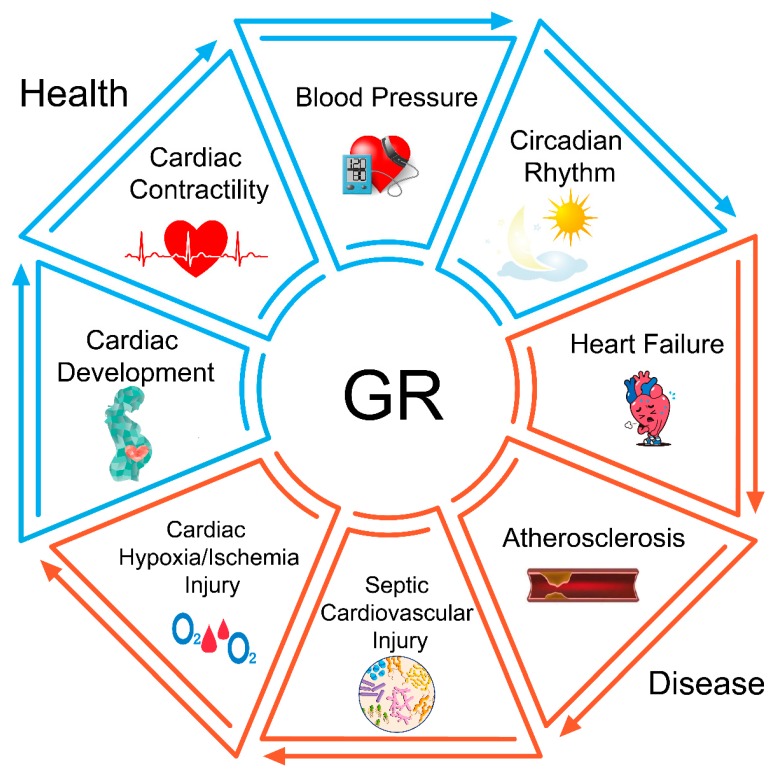
GR influences a variety of physiological processes in states of health and disease.

**Table 1 cells-08-01227-t001:** Polymorphisms in the NR3C1 gene related to cardiovascular diseases (CVDs).

Polymorphism	Glucocorticoid Sensitivity	Involved Risk Factors/Disease	References
ER22/23EK	Decreased	Lower risk of type 2 diabetes mellitus and cardiovascular disease	[50]
N363S	Increased	Obesity, type 2 diabetes, coronary artery disease	[54,55,56,57,58]
GR-9β	Decreased	Decreased total cholesterol levels and increased HDL cholesterol levels, regulation of human blood pressure	[59]
BclI	Increased	Hypertension, adiposity, obesity, atherosclerosis	[60,61,62,63,64,65,66,67]
D401H	Increased	Hypertension, diabetes, accumulation of visceral fat	[68]
A714Q	Decreased	Hypoglycemia, hypertension	[69]
F737L	Decreased	Hypertension, hypokalemia	[70]
F774S	Decreased	Hypoglycemia, hypertension	[71]
V575G	Decreased	Hypertension, hypokalemia	[72]
D641V	Decreased	Hypertension, hypokalemic alkalosis	[73]
G679S	Decreased	Hypertension, fatigue	[74]

**Table 2 cells-08-01227-t002:** The summary of GR in cardiac development.

Animal Model	Study Type	GR Knock Out Condition	Outcomes	Reference
Mouse	In vivo	Global	Mice died soon after birth because of organ dysfunction.	[76]
Mouse	In vivo	Heart-specific	Mice died prematurely from pathological cardiac hypertrophy.	[78]
Mouse	In vitro	Heart-specific	Expression of several key genes regarding cardiac contractility, cardiomyocyte survival, and inflammation changed.	[78]
Piglet	In vitro	N/A	GR-related pathways that participated in the regulation of myocyte size.	[80]
Piglet	In vitro	N/A	GR-related pathways were related with myocyte structural maturation.	[81]

N/A: not available.

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
