# Peer review of "The Glucocorticoid Receptor in Cardiovascular Health and Disease"

_cells, 2019, doi:10.3390/cells8101227_

Round 1

Reviewer 1 Report

The authors have prepared a comprehensive review and update to the role of glucocorticoid receptor on cardiovascular health and disease.  The narrative presented and supporting literature is appropriate. I compliment the authors on their work.

Author Response

We thank you for your complimentary remarks.

Reviewer 2 Report

In this excellent review, Liu and colleague carefully and comprehensively analyzed the glucocorticoid (GC) literature and specifically focused on the cardiovascular system. After providing background information about the GC receptor (GR), its actions and metabolism, the authors described in details the role GR in the cardiovascular system both in physiological (e.g. cardiac development and contractility, regulation of the blood pressure) and pathological conditions (e.g. heath failure, atherosclerosis, hypoxia/ischemia). Overall, this review manuscript is well-written and clear. However, it contains some minor issues that need the authors’ attention.

Minor comments:

Line 77: The authors should elaborate further on how co-regulators interacts with AF2 upon ligand binding? From timing standpoint, which region interacts first with co-regulators upon ligand binding? Is it AF1 or AF2?

Line 82: The authors stated that GR has 2 isoforms. However, GR has more than 2 isoforms. Correction is needed.

Line 88: The authors should also elaborate further on the differences in terms of the regulation of gene expression between the multiple GR-alpha translational isoforms. While authors stated that those isoforms have similar ability to bind to GREs, some differences have been reported by Cidlowski’s group in terms of function (Mol Cell Biol. 2007 Oct;27(20):7143-60).

Line 102: The authors should elaborate a little more about the GR-beta functions beyond its dominant negative activities toward GR-alpha.

Line 103: The use of the term “exert” here is not clear.

Line 103  – Line 110: Section related to other GR isoforms is not clear and needs to be rewritten and elaborated more.

Line 118: Correct “conisists”

Line 120: The role GR phosphorylation should be briefly mentioned here.

Line 139: Correct “containsbinding”.

Page 5 - Fig. 2: This figure is confusing: Is the GR complex (bound to other chaperons) that translocate to the nucleus and the free GR exert non-genomic effects?

Line 155: The term “peripheral resistance” is not clear.

Line 166: Should be Fig. 2.

Line 181-182: Sentence is not clear and needs to be rewritten and/or elaborated further.

Line 221: The table contains additional mutations not commented in the text. They should be at least briefly stated.

Section 6.1: This section may need its own table to enhance clarity.

Line 245: The suggestion that “there is selective GR” is not clear and should be further justified.

Line 463: GC is known to reduce GR expression (homologous downregulation) (Crit Rev Eukaryot Gene Expr. 1993;3(2):63-88.). The statement that DEX increases the expression of GR needs to be discussed in the context of glucocorticoid-induced GR homologous downreguation.

Page 14 - Conclusion and Future Directions:  I would suggest to present the future directions as bullet points to enhance clarity.

The text contains few minor typos that need the authors’ attention.
